# Proteome-wide signatures of function in highly diverged intrinsically disordered regions

**Taraneh Zarin[1], Bob Strome[1], Alex N Nguyen Ba[2], Simon Alberti[3,4], Julie D Forman-Kay[5,6], Alan M Moses[1,7,8]\***

[1]Department of Cell and Systems Biology, University of Toronto, Toronto, Canada; [2]Department of Organismic and Evolutionary Biology, Harvard University, Cambridge, United States; [3]Max Planck Institute of Molecular Cell Biology and Genetics, Dresden, Germany; [4]Center for Molecular and Cellular Bioengineering, Biotechnology Center, Technische Universität Dresden, Dresden, Germany; [5]Program in Molecular Medicine, Hospital for Sick Children, Toronto, Canada; [6]Department of Biochemistry, University of Toronto, Toronto, Canada; [7]Department of Ecology and Evolutionary Biology, University of Toronto, Toronto, Canada; [8]Department of Computer Science, University of Toronto, Toronto, Canada

**Abstract** Intrinsically disordered regions make up a large part of the proteome, but the sequence-to-function relationship in these regions is poorly understood, in part because the primary amino acid sequences of these regions are poorly conserved in alignments. Here we use an evolutionary approach to detect molecular features that are preserved in the amino acid sequences of orthologous intrinsically disordered regions. We find that most disordered regions contain multiple molecular features that are preserved, and we define these as 'evolutionary signatures' of disordered regions. We demonstrate that intrinsically disordered regions with similar evolutionary signatures can rescue function in vivo, and that groups of intrinsically disordered regions with similar evolutionary signatures are strongly enriched for functional annotations and phenotypes. We propose that evolutionary signatures can be used to predict function for many disordered regions from their amino acid sequences.

**\*For correspondence:**
alan.moses@utoronto.ca

**Competing interests:** The authors declare that no competing interests exist.

## Introduction

Intrinsically disordered protein regions are associated with a large array of functions (reviewed in *Forman-Kay and Mittag, 2013*), including cell signaling (*Iakoucheva et al., 2004*; *Tompa, 2014*; *Wright and Dyson, 2015*), mediation of protein-protein interactions (*Borgia et al., 2018*; *Tang et al., 2012*; *Tompa et al., 2015*), and the formation of membraneless organelles through phase separation (*Banani et al., 2017*; *Franzmann et al., 2018*; *Nott et al., 2015*; *Patel et al., 2015*; *Riback et al., 2017*). These regions are widespread in eukaryotic proteomes (*Peng et al., 2013*; *Ward et al., 2004*), but do not fold into stable secondary or tertiary structures, and do not typically perform enzymatic functions (*Uversky, 2011*). Although intrinsically disordered regions can readily be identified based on their primary amino acid sequence (*Dosztányi et al., 2005*; *Uversky, 2002*), it remains a challenge to associate these regions with specific biological and biochemical functions based on their amino acid sequences, limiting systematic functional analysis. In stark contrast, for folded regions, protein function can often be predicted with high specificity based on the presence of conserved protein domains (*El-Gebali et al., 2019*) or enzymatic active sites (*Ondrechen et al., 2001*). Analogous methods to assign function to intrinsically disordered regions

based on evolutionary conservation (or other sequence properties) are of continuing research interest (reviewed in *van der Lee et al., 2014*).

We and others (*Davey et al., 2012*; *Nguyen Ba et al., 2012*) have shown that short segments of evolutionary conservation in otherwise rapidly evolving disordered regions point to key functional residues, often important for posttranslational modifications, or other transient protein interactions (*Tompa et al., 2014*). However, these conserved segments make up a small fraction of disordered regions (5%), and the vast majority of disordered amino acids show little evidence for evolutionary constraint in alignments of primary amino acid sequences (*Colak et al., 2013*). It is currently unclear how intrinsically disordered regions persist at high frequency in the proteome, given these apparently low levels of evolutionary constraint (*Afanasyeva et al., 2018*; *Brown et al., 2010*; *Colak et al., 2013*; *de la Chaux et al., 2007*; *Khan et al., 2015*; *Light et al., 2013*; *Tóth-Petróczy and Tawfik, 2013*).

One hypothesis for the preponderance of disordered regions despite high amino acid sequence divergence, is that the 'molecular features' of disordered regions that are important for function (such as length [*Schlessinger et al., 2011*], complexity [*Alberti et al., 2009*; *Halfmann, 2016*; *Kato et al., 2012*; *Molliex et al., 2015*], amino acid composition [*Moesa et al., 2012*], and net charge [*Mao et al., 2010*; *Strickfaden et al., 2007*; *Zarin et al., 2017*]) do not lead to detectable similarity in primary amino acid sequence alignments. Indeed, recently, evidence that such molecular features can be under evolutionary constraint has been reported for some proteins (*Daughdrill et al., 2007*; *Lemas et al., 2016*; *Zarin et al., 2017*). For example, we showed that signaling function of a disordered region in the *Saccharomyces cerevisiae* protein Ste50 appears to depend on its net charge, and we found evidence that this molecular feature is under evolutionary constraint, despite no detectable sequence similarity in alignments (*Zarin et al., 2017*).

Here we sought to test whether evolutionary preservation of molecular features is a general property of highly diverged intrinsically disordered protein regions. To do so, we obtained a set of 82 sequence features reported in the literature to be important for disordered region function (*Supplementary file 1* - Table S1). We computed these for *S. cerevisiae* intrinsically disordered regions and their orthologs, and compared them to simulations of molecular evolution where conserved segments (if any) are retained, but where there is no selection to retain molecular features (*Nguyen Ba et al., 2014*; *Nguyen Ba et al., 2012*). Deviations from the simulations indicate that the highly diverged intrinsically disordered regions are preserving molecular features during evolution through natural selection (*Zarin et al., 2017*).

We find that many intrinsically disordered regions show evidence for selection on multiple molecular features, which we refer to as an 'evolutionary signature'. Remarkably, we show that intrinsically disordered regions with similar evolutionary signatures appear to rescue function, while regions with very different signatures cannot, strongly supporting the idea that the preserved molecular features are important for disordered region function. By clustering intrinsically disordered regions based on these evolutionary signatures, we obtain (to our knowledge) the first global view of the functional landscape of these enigmatic protein regions. We recover patterns of molecular features known to be associated with intrinsically disordered region functions such as subcellular organization and targeting signals. We also identify new patterns of molecular features not previously associated with functions of disordered regions such as DNA repair and ribosome biogenesis. Finally, we show that similarity of evolutionary signatures can generate hypotheses about the function of completely disordered proteins. Taken together, our results indicate that evolutionary constraint on molecular features in disordered regions is so widespread that sequence-based prediction of their functions should be possible based on molecular features.

## Results

### Proteome-wide evolutionary analysis reveals evolutionarily constrained sequence features are widespread in highly diverged intrinsically disordered regions

We identified more than 5000 intrinsically disordered regions (IDRs) in the *S. cerevisiae* proteome and quantified their evolutionary divergence (see Materials and methods). As expected, we found that the IDRs evolve more rapidly than the regions that were not identified as disordered (*Figure 1—*

*figure supplement 1*). We also confirmed that the vast majority of these IDRs are distinct from Pfam domains (*Figure 1—figure supplement 2*). These results are consistent with previous reports (*Brown et al., 2010*; *Colak et al., 2013*; *de la Chaux et al., 2007*; *Khan et al., 2015*; *Light et al., 2013*; *Tóth-Petróczy and Tawfik, 2013*) that the primary amino acid sequence alignments of IDRs show high levels of divergence and it is not possible to annotate IDR functions using standard homology-based approaches.

To test for selection on molecular features in these IDRs, we applied a method that we recently used to show evidence of selection on an IDR in the *S. cerevisiae* Ste50 protein (*Zarin et al., 2017*). We obtained 82 molecular features that have been reported or hypothesized to be important for IDR function (*Supplementary file 1* - Table S1) and tested whether these molecular features are under selection in the *S. cerevisiae* IDRs (see Materials and methods for details). Briefly, we compare the distribution of a given molecular feature in a set of orthologous IDRs to a null expectation, which is formed by simulating the evolution of each IDR. When the mean or variance of the molecular feature across the orthologous IDRs deviates from the distribution of means or variances in our null expectation, we predict that this feature is under selection, and thus could be important for the function of the IDR in question. For example, in the Ste50 IDR, as reported previously (*Zarin et al., 2017*), we found that the variance of the net charge with phosphorylation of the IDR falls outside of our null expectation, while the mean falls within our null expectation (*Figure 1A*).

We applied this analysis to 5149 IDRs (see Materials and methods) and computed the percentage of IDRs where the evolution of each molecular feature fell beyond our null expectation (empirical $p < 0.01$, *Figure 1B*). We find that charge properties such as net charge and acidic residue content are most likely to deviate from our null expectation (more than 50% of IDRs) (*Figure 1B*). This is in contrast to non-conserved motif density, which deviates from our null expectation in 21.6% of IDRs at most (for CDK phosphorylation consensus sites). Other molecular features that frequently deviate from our null expectation are sequence complexity (43.0%), asparagine residue content (43.3%), and physicochemical features such as isoelectric point (53.9%). We also found that the mean of each molecular feature deviates from our null expectation more often than the variance (*Figure 1B*). These results suggest that there are many more molecular features that are under selection in IDRs than is currently appreciated (*Daughdrill et al., 2007*; *Lemas et al., 2016*; *Zarin et al., 2017*).

Next, we quantified the number of molecular features that are significant per IDR, assigning significance to a molecular feature if either the mean, variance, or both mean and variance of the molecular feature deviated from our null expectation (empirical $p < 0.01$, *Figure 1C*). Surprisingly, many IDRs have many significant molecular features, with a median of 15 significant molecular features per IDR (compared to one significant feature expected by chance; see Materials and methods). Although many of our features are correlated (see Discussion), these results suggest that the deviation from our expectations of molecular feature evolution is not due to a few outlier IDRs, but rather that most IDRs tend to have multiple molecular features that are under selection.

## Intrinsically disordered regions with similar molecular features can perform similar functions despite negligible similarity of primary amino acid sequences

The analysis above indicates that highly diverged IDRs typically contain multiple molecular features that are under selection. To summarize the set of preserved molecular features in each IDR, we computed Z-scores comparing either the observed mean or variance of each molecular feature in the orthologous IDRs to our simulations (see Materials and methods). We call these summaries of evolution of molecular features (vectors of Z-scores) 'evolutionary signatures'. If the features are important for function, IDRs with similar evolutionary signatures are predicted to perform (or at least be capable of performing) similar molecular functions. To test this hypothesis, we replaced the endogenous Ste50 IDR with several IDRs from functionally unrelated proteins: Pex5, a peroxisomal signal receptor (*Erdmann and Blobel, 1996*), Stp4, a predicted transcription factor (*Abdel-Sater et al., 2004*), and Rad26, a DNA-dependent ATPase involved in Transcription Coupled Repair (*Gregory and Sweder, 2001*; *Guzder et al., 1996*) (*Figure 2A*). Ste50 is an adaptor protein in the High Osmolarity Glycerol (HOG) and mating pathways (*Hao et al., 2008*; *Jansen et al., 2001*; *Tatebayashi et al., 2007*; *Truckses et al., 2006*; *Yamamoto et al., 2010*) whose IDR is important for basal mating pathway activity (as measured by expression of a reporter driven by the Fus1 promoter) (*Hao et al., 2008*; *Zarin et al., 2017*). The IDRs that we used to replace the Ste50 IDR all have negligible similarity

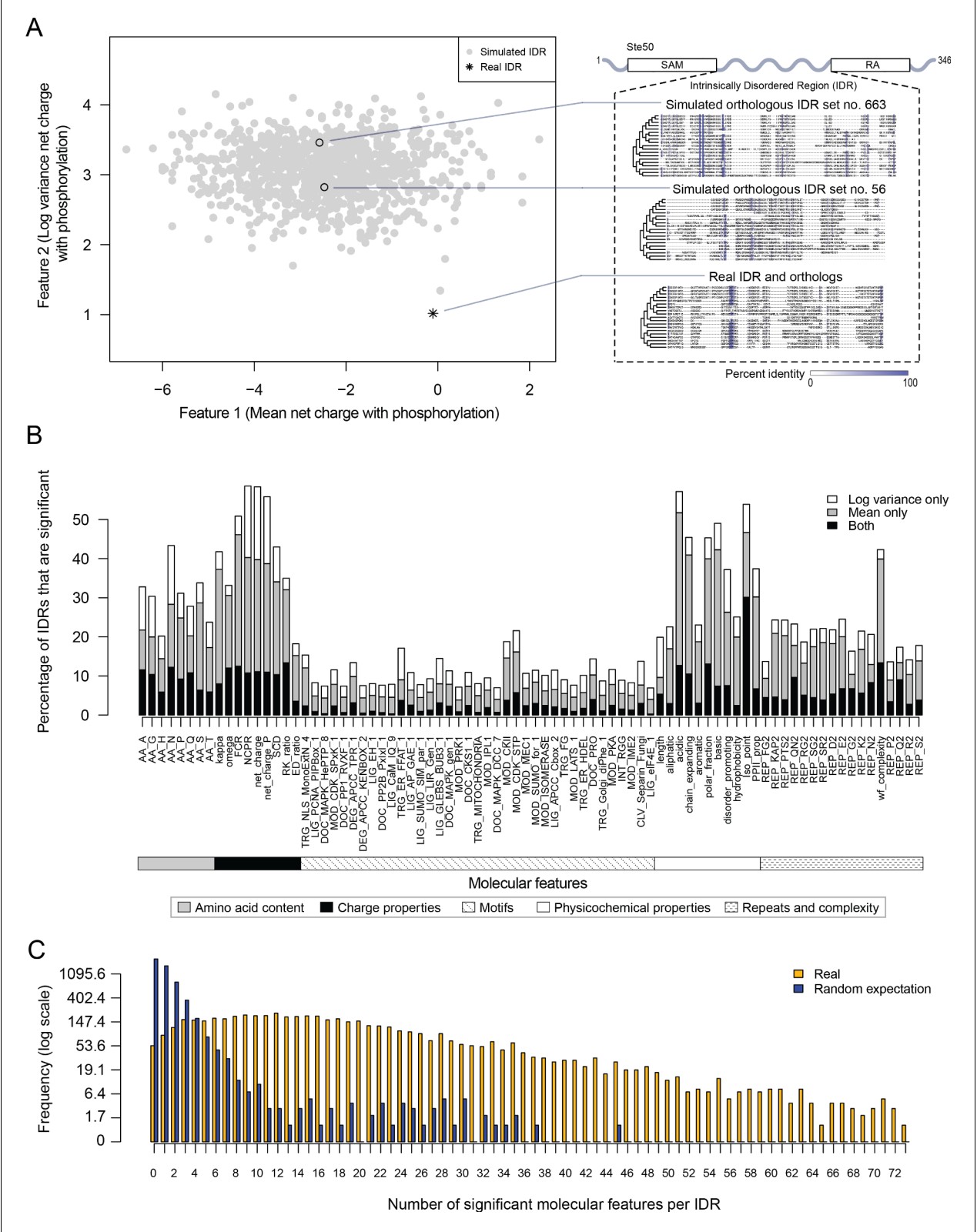

**Figure 1.** Proteome-wide evolutionary analysis reveals evolutionarily constrained sequence features are widespread in highly diverged intrinsically disordered regions. (**A**) Left: Mean versus log variance of the 'net charge with phosphorylation' molecular feature for the real Ste50 IDR (a.a. 152–250) ortholog set and simulated Ste50 orthologous IDR sets (N = 1000). Right: Example simulated Ste50 orthologous IDR sets (no. 663 and no. 56 out of 1000) and the real Ste50 IDR and its orthologs, colored according to percent identity in the primary amino acid sequence. (**B**) Percentage of IDRs that

*Figure 1 continued on next page*

Figure 1 continued

are significantly deviating from simulations in mean, log variance, or both mean and log variance of each molecular feature. (C) Frequency [1 + log (frequency)] of number of significant molecular features per IDR for the real IDRs (yellow) versus the random expectation (blue) obtained from a set of simulated IDRs.

The online version of this article includes the following figure supplement(s) for figure 1:

**Figure supplement 1.** Predicted IDRs in the *S. cerevisiae* proteome ('IDR') are more highly diverged compared to regions that are not predicted to be disordered ('non-IDR') (p<2.2×10$^{-16}$, Wilcoxon test).

**Figure supplement 2.** Percentage of overlap with Pfam domains for IDRs predicted to be disordered in the *S. cerevisiae* proteome that are >= 30 amino acids ('IDR') have less overlap with Pfam domains compared to all other regions that are >= 30 amino acids ('non-IDR') (p<2.2 × 10$^{-16}$, Wilcoxon test).

when their primary amino acid sequences are aligned, but vary in the similarity of their evolutionary signatures (to the Ste50 IDR, *Figure 2A*). We found that the basal mating reporter expression in each strain corresponded to how similar the evolutionary signature of the replacing IDR was to that of the Ste50 IDR (all mutants significantly different from wildtype and each other, Wilcoxon test p<0.05, *Figure 2B*). To further assay mating pathway activity, we exposed the wildtype and chimaeric strains with IDRs from Pex5, Stp4 and Rad26 to mating pheromone. We found that the two chimaeric strains that were more similar in their evolutionary signatures to the wildtype (Pex5 and Stp4) began the process of 'shmooing', or responding to pheromone, whereas the strain that had the IDR with the most different evolutionary signature (Rad 26) could not shmoo (*Figure 2C*; full micrographs in *Figure 2—figure supplement 1*). Although it is challenging to quantitate similarity between evolutionary signatures comprised of correlated features (see Discussion), our ability to use these signatures to predict which IDRs can rescue signaling function suggests that they may be associated with IDR function.

## Proteome-wide view of evolutionary signatures in disordered regions reveals association with function

To test the association of function with evolutionary signatures in highly diverged IDRs, we clustered and visualized the evolutionary signatures for 4646 IDRs in the proteome (see Materials and methods) (*Figure 3*). Remarkably, the evolutionary signatures reveal a global view of disordered region function. The IDRs fall into at least 23 clusters based on similarity of their evolutionary signatures (groups A through W, *Figure 3*) that are significantly associated with specific biological functions (enriched for Gene Ontology (GO) term, phenotype, and/or literature annotations, False Discovery Rate [FDR]=5%, Benjamini-Hochberg corrected) (*Table 1*; full table of enrichments in supplementary data; clustered IDRs and evolutionary signatures in supplementary data). Given that this level of specificity of biological information has not been previously associated with sequence properties of highly diverged IDRs, we performed a series of controls, ensuring that our clusters are not based on sequence similarity between IDRs, that our annotation enrichment results are not due to a mis-specification of the null hypothesis, and to confirm that these annotation enrichment results cannot be obtained simply based on amino acid frequencies of IDRs (*Supplementary file 1* - Table S2; see Materials and methods).

Several of the functions that we find enriched within our clusters have been previously associated with molecular features of IDRs, which we recover in our analysis. For example, we find a cluster that is associated with 'nucleocytoplasmic transporter activity' (cluster M) that includes IDRs from FG-NUP proteins Nup42, Nup145, Nup57, Nup49, Nup116, and Nup100 that form part of the nuclear pore central transport channel (*Alber et al., 2007*). In cluster M, we find molecular features such as increased asparagine content, increased polar residue content, and increased proline and charged residue demixing ('Omega'; *Martin et al., 2016*) in addition to the well-known 'FG' repeats that are found in the FG-NUP IDRs (reviewed in *Terry and Wente, 2009*). Another interesting example is cluster O, which contains IDRs from proteins that are enriched for a wide range of annotations such as 'P-body', 'cytoplasmic stress granule', 'actin cortical patch', and 'DNA binding'. Cluster O contains IDRs from proteins associated with phase separation and membraneless organelles such as Sup35 (*Franzmann et al., 2018*) and Dhh1 (*Protter et al., 2018*). The evolutionary signatures for the IDRs in this cluster include features that are typically associated with so-called 'prionogenic', low complexity disordered regions, such as increased mean polyglutamine repeats (*Alberti et al., 2009*),

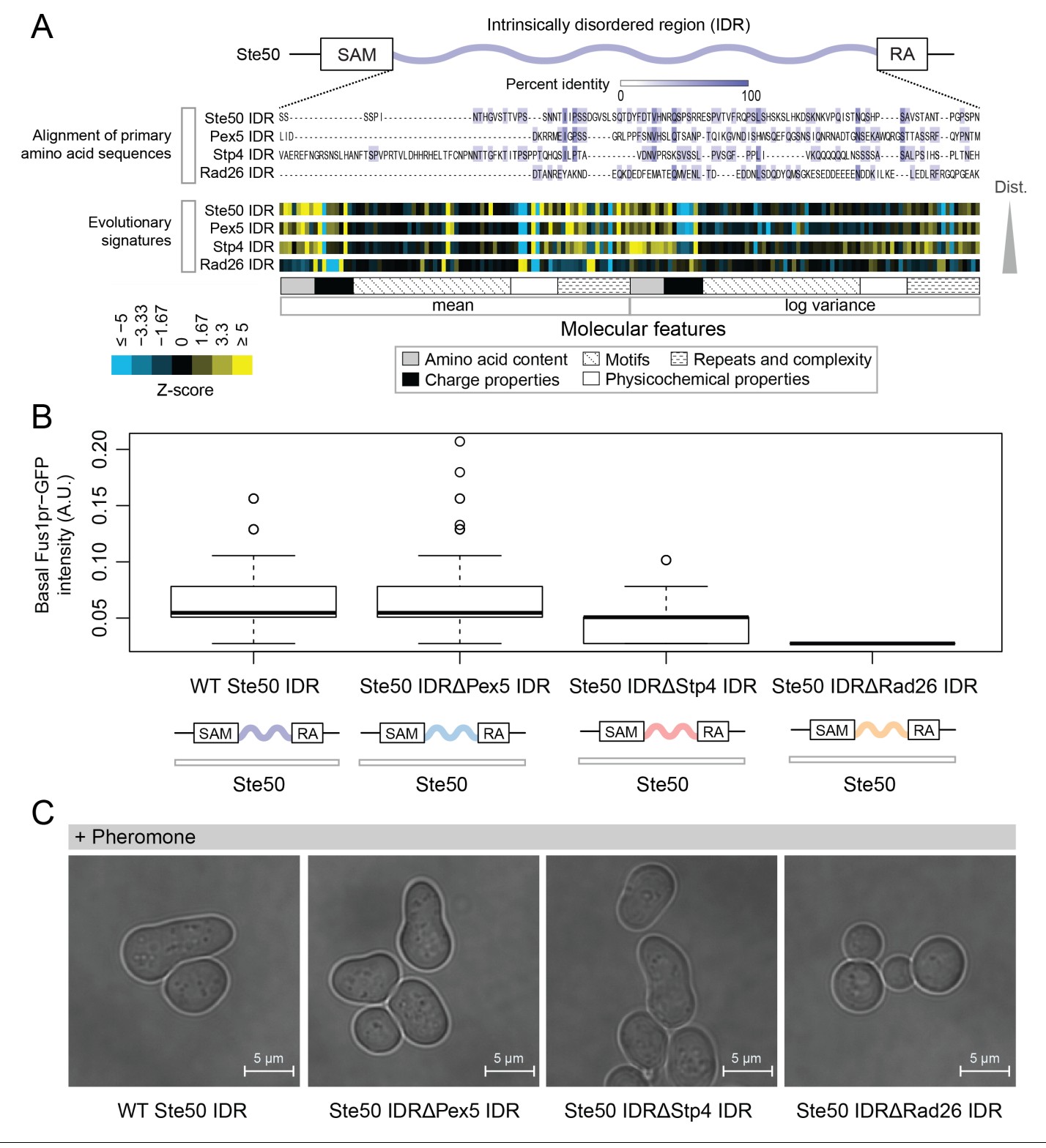

**Figure 2.** Intrinsically disordered regions with similar evolutionary signatures can rescue wildtype phenotypes, while those with different evolutionary signatures cannot. (**A**) Multiple sequence alignment of Ste50 IDR (a.a. 152–250), Pex5 IDR (a.a. 77–161), Stp4 (a.a. 144–256), and Rad26 IDR (a.a. 163–239) shows negligible similarity when their primary amino acid sequences are aligned, while evolutionary signatures show that the Pex5 and Stp4 IDRs are more similar to the Ste50 IDR than the Rad26 IDR. While the Ste50 IDR has five consensus phosphorylation sites that are implicated in its function (*Hao et al., 2008*; *Yamamoto et al., 2010*; *Zarin et al., 2017*), the Pex5 IDR and Rad26 IDR have none, and the Stp4 IDR has 4. IDRs are presented in

*Figure 2 continued on next page*

*Figure 2 continued*

order of increasing Euclidian distance between their evolutionary signatures, though we do not recommend using this measure to quantitate similarity between evolutionary signatures independently (see Discussion). The Ste50 IDR is located between the Sterile Alpha Motif (SAM) and Ras Association (RA) domains in the Ste50 protein. (**B**) Boxplots show distribution of values corresponding to basal Fus1pr-GFP activity in an *S. cerevisiae* strain with the wildtype Ste50 IDR compared to strains with the Pex5, Stp4, or Rad26 IDR swapped to replace the Ste50 IDR in the genome. Boxplot boxes represent the 25th-75th percentile of the data, the black line represents the median, and whiskers represent 1.5*the interquartile range. Outliers fall outside the 1.5*interquartile range, and are represented by unfilled circles. Distribution of GFP activity is based on quantification of GFP intensity in single cells pooled from four colonies (which we define as biological replicates) for each strain; sample sizes for each distribution are as follows: WT n = 588 cells, Pex5 IDR n = 196 cells, Stp4 IDR n = 228 cells, Rad26 IDR n = 271 cells. (**C**) Brightfield micrographs showing each strain from part B following exposure to pheromone. Shmooing cells are those which have elongated cell shape, representing mating projections.

The online version of this article includes the following figure supplement(s) for figure 2:

**Figure supplement 1.** Full field-of-view micrographs of pheromone-exposed *S. cerevisiae* strains from *Figure 2C*.

but also indicate that there are other relevant molecular features for this set of disordered regions (*Figure 4A*). For example, in these regions, the variance of the net charge is reduced, and charged residues are depleted during evolution. These sequence features are illustrated in *Figure 4B*, where we compare the presence of glutamine and charged residues in an example disordered region from this cluster (Ccr4; a protein that is known to accumulate in P-bodies; *Teixeira and Parker, 2007*) to an example from the corresponding simulation (*Figure 4B*). Taken together, these results indicate that our analysis captures molecular features that have been previously associated with IDR functions, and suggests additional molecular features in these IDRs that may be important for their functions.

We also find functions associated with our clusters that have not been previously associated with molecular features of IDRs. For example, cluster D (*Figure 5A*) is associated with DNA repair, and its evolutionary signature contains increased mean 'Kappa' (*Das and Pappu, 2013*) and decreased mean 'Sequence Charge Decoration' (SCD) (*Sawle and Ghosh, 2015*), both of which indicate that there is an increased separation of positive and negatively charged residues in these IDRs compared to our null expectation. This is illustrated by the IDR from Srs2, a protein that is known to be involved in DNA repair (*Aboussekhra et al., 1989*; *Yeung and Durocher, 2011*), and shows high charge separation compared to an example corresponding simulation (*Figure 5B*). The evolutionary signature for this cluster also reveals an increased mean fraction of charged residues and negatively charged residues in particular (*Figure 5A*), which is also clear in the comparison between the real Srs2 orthologs and the simulation (*Figure 5B*). Although acidic stretches have been associated with IDRs in histone chaperones (*Warren and Shechter, 2017*), to our knowledge, the separation of oppositely charged residues has not been associated with the wider functional class of DNA repair IDRs.

Our analysis also indicates that there is not necessarily a 1:1 mapping between IDRs with shared evolutionary signatures and current protein functional annotations. For example, we find three clusters associated with ribosome biogenesis (clusters A, C, F) that cannot be distinguished based on their enriched GO terms. The largest of these is cluster A, where 201/295 proteins have a 'nucleus' annotation, and 110/295 are essential proteins ('inviable' deletion phenotype). This cluster is also enriched for several phenotypes associated with RNA accumulation (*Table 1*, cluster A; see supplementary data for full list of significant enrichments). Cluster A contains highly acidic IDRs with CKII phosphorylation consensus sites. CKII has been previously associated with nucleolar organization (*Louvet et al., 2006*), and a previous analysis of non-conserved consensus phosphorylation sites found ribosome biogenesis as strongly enriched in predicted CKII targets (*Lai et al., 2012*). In contrast, cluster C shares neither of these molecular features with cluster A, and cluster F shares only highly acidic residue content. Interestingly, cluster C contains increased mean polylysine repeats, and is significantly enriched for proteins that have been experimentally verified as targets for lysine polyphosphorylation (*Bentley-DeSousa et al., 2018*) (p=$2.7 \times 10^{-3}$, hypergeometric test). Overall, although the IDRs in these clusters share different evolutionary signatures, they are all found in proteins associated with ribosome biogenesis. We hypothesize that these different signatures point to different functions relating to ribosome biogenesis, but we have no indication of what these might be based on current protein annotations (see Discussion).

We find similar observations in multiple clusters that have distinct evolutionary signatures enriched for terms associated with regulation of transcription (clusters I, J, L, N, O, R). These clusters

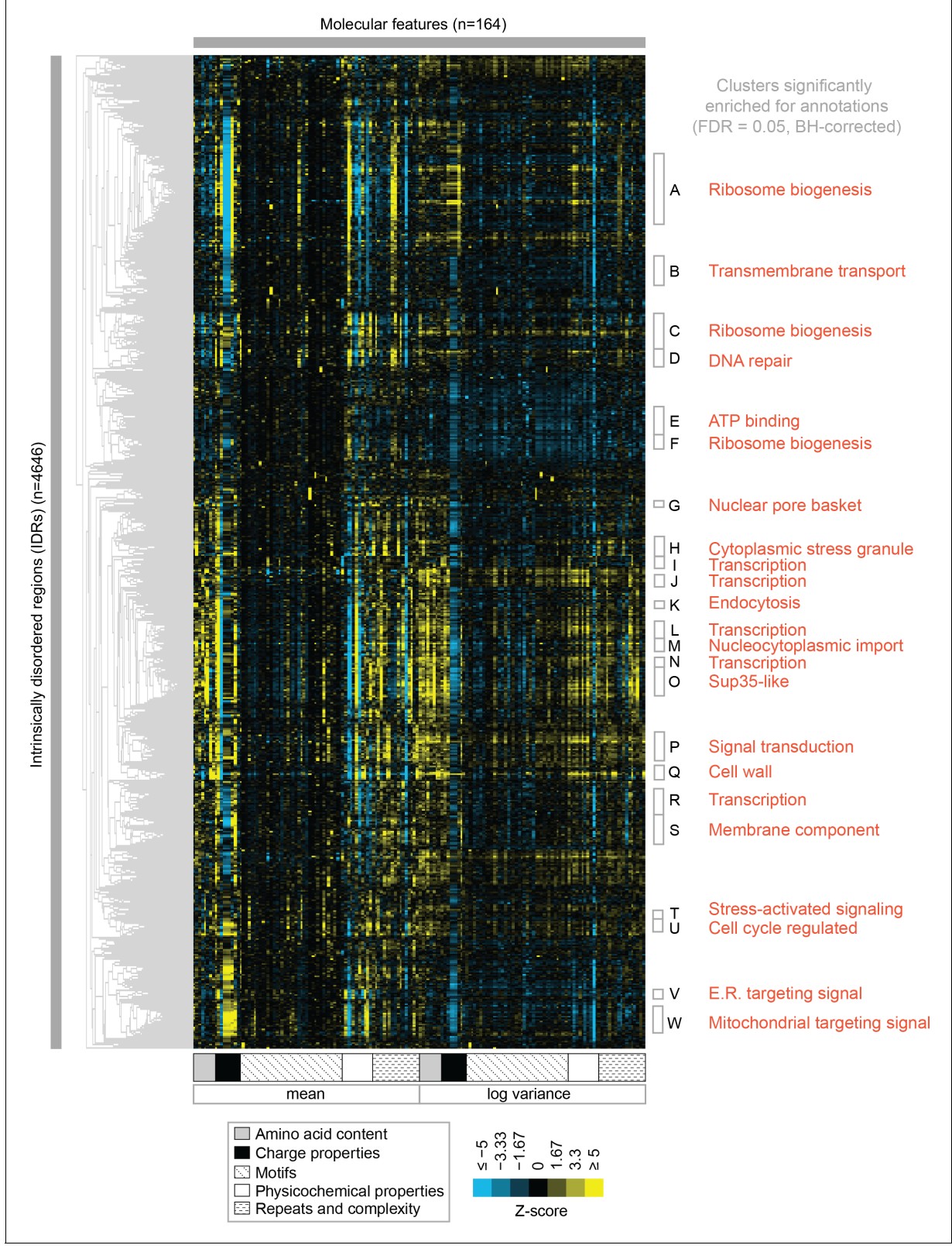

**Figure 3.** Clustering evolutionary signatures shows that IDRs in the proteome share evolutionary signatures, and that these clusters of IDRs are associated with specific biological functions. A-W show clusters significantly enriched for annotations (see *Table 1*; full table of enrichments in supplementary data). Cluster names represent summary of enriched annotations.

*Figure 3 continued on next page*

*Figure 3 continued*

The online version of this article includes the following source data for figure 3:

**Source data 1.** Clustered IDR data and mapping between IDRs and Cluster ID assigned in *Figure 3*.

are not clearly separable based on mechanistic steps of transcription (such as sequence-specific DNA binding, chromatin remodeling, etc.). Some of these clusters exhibit molecular features that have been associated with different classes of transcriptional activation domains that are based on amino acid composition (reviewed in *Frietze and Farnham, 2011*). For example, clusters J, O and N have increased glutamine residue content, and cluster N has increased proline residue content. However, clusters I and R have no amino acid composition bias, while cluster N has increased proline-directed phosphorylation consensus sites, suggesting post-translational modifications. This indicates that our analysis reveals new sub-classifications of transcription-associated IDRs. While we hypothesize that these IDRs have different functions, once more we have no indication of what these functions could be based on current protein annotations (see Discussion).

## A cluster of evolutionary signatures is associated with N-terminal mitochondrial targeting signals

One of our clusters of intrinsically disordered regions is exceptionally strongly associated with the mitochondrion (144/165 proteins in the cluster) and other annotations that are related to mitochondrial localization and function (for example, 81/165 proteins in the cluster have shown a decreased respiratory growth phenotype) (*Table 1*, cluster W; see supplementary data for full list of significant enrichments). The vast majority of mitochondrial proteins are synthesized with N-terminal pre-sequences (*Maccecchini et al., 1979*) (also known as N-terminal targeting signals) that are cleaved upon import (*Vögtle et al., 2009*) and are thought to sample dynamic structural configurations (*Saitoh et al., 2011*; *Saitoh et al., 2007*) (*Figure 6A*). Since 145/165 of the disordered regions in this cluster are N-terminal, we hypothesized that this cluster contains disordered regions that are associated with mitochondrial targeting signals (*Vögtle et al., 2009*). In line with this hypothesis, we find previously described sequence features of mitochondrial N-terminal targeting signals in our evolutionary signatures; for example, these IDRs are depleted of negatively charged residues, have an abundance of positively charged residues, and are much more hydrophobic than our null expectation (*Figure 6—figure supplement 1A*) (*Garg and Gould, 2016*; *Vögtle et al., 2009*). Examples of disordered regions in this cluster include those of the Heme A synthase Cox15 and the mitochondrial inner membrane ABC (ATP-binding cassette) transporter Atm1 (*Figure 6—figure supplement 1B*). In order to test our hypothesis that this cluster of evolutionary signatures identifies mitochondrial N-terminal targeting signals, we used a recently published tool that scores the probability that a sequence is a mitochondrial targeting signal (*Fukasawa et al., 2015*). Using this tool, we find that the IDRs in cluster W have a much higher probability of being mitochondrial targeting signals than any other cluster with enriched annotations in our analysis (Bonferroni-corrected $p \leq 6.5 \times 10^{-11}$, Wilcoxon test) (*Figure 6B*, red box). Interestingly, the adjacent cluster V (*Figure 6B*, purple box), which we hypothesize to contain targeting sequences for the endoplasmic reticulum, is distinct from cluster W in this analysis.

If the specificity of the function of the IDRs in this cluster is strong, we predict that swapping an IDR from cluster W with that of a verified mitochondrial targeting sequence would result in correct localization to the mitochondria, while swapping an IDR from a different cluster would not. Mitochondrial precursor peptides that are not correctly targeted to the mitochondria are thought to be degraded (*Mårtensson et al., 2019*; *Weidberg and Amon, 2018*). To test this, we first used the (uncharacterized) disordered region from Atm1 that falls into cluster W to replace that of Cox15, which also falls into cluster W and is an experimentally verified mitochondrial targeting sequence (*Vögtle et al., 2009*) (*Figure 6C*). In accordance with our hypothesis, we find that GFP-tagged Cox15 correctly localizes to the mitochondria when its disordered region is swapped with that of Atm1, but does not localize correctly when its disordered region is deleted (*Figure 6C*; full micrographs in *Figure 6—figure supplement 2*). We also repeated this experiment with another protein that has an experimentally verified N-terminal mitochondrial targeting sequence, Mdl2, and found the same results (*Figure 6—figure supplement 3*). Next, we replaced the Cox15 IDR with the

**Table 1.** Top five enriched GO term annotations and top three enriched phenotype annotations for each cluster (in order of decreasing corrected p-values).

Full table of >1000 significant GO term, phenotype, and literature enrichments in supplementary data.

| ID | Annotations (Positive proteins in cluster/Total proteins in cluster) | Corrected P <= |
|---|---|---|
| A | nucleus (201/295), rRNA processing (40/295), ribosome biogenesis (39/295), nucleolus (50/295), maturation of SSU-rRNA from tricistronic rRNA transcript (SSU-rRNA, 5.8S rRNA, LSU-rRNA) (14/295), inviable (110/295), RNA accumulation decreased (46/295), RNA accumulation increased (39/295) | 1.46e-03 |
| B | amino acid transmembrane transport (8/140), amino acid transmembrane transporter activity (8/140), transmembrane transport (21/140), amino acid transport (9/140) | 1.11e-02 |
| C | nucleolus (42/159), rRNA processing (27/159), ribosome biogenesis (26/159), nucleus (107/159), preribosome, large subunit precursor (13/159), RNA accumulation increased (28/159), inviable (60/159), RNA accumulation decreased (27/159) | 4.88e-03 |
| D | nucleus (72/86), DNA repair (20/86), cellular response to DNA damage stimulus (18/86), DNA binding (28/86), damaged DNA binding (7/86), mutation frequency increased (14/86), chromosome plasmid maintenance decreased (29/86), cell cycle progression in S phase increased duration (4/86) | 4.21e-02 |
| E | motor activity (4/89), ATP binding (25/89), ASTRA complex (3/89) | 4.23e-02 |
| F | 90S preribosome (11/73), rRNA processing (14/73), ribosome biogenesis (14/73), endonucleolytic cleavage in ITS1 to separate SSU-rRNA from 5.8S rRNA and LSU-rRNA from tricistronic rRNA transcript (SSU-rRNA, 5.8S rRNA, LSU-rRNA) (6/73), nucleolus (15/73) | 2.49e-02 |
| G | nuclear pore nuclear basket (4/35), nucleocytoplasmic transporter activity (4/35) | 4.54e-02 |
| H | nucleic acid binding (16/66), translational initiation (7/66), cytoplasmic stress granule (9/66), mRNA binding (13/66), translation initiation factor activity (6/66) | 3.60e-03 |
| I | regulation of transcription, DNA-templated (23/52), transcription, DNA-templated (22/52), positive regulation of transcription from RNA polymerase II promoter (12/52) | 6.58e-03 |
| J | RNA polymerase II transcription factor activity, sequence-specific DNA binding (10/52), positive regulation of transcription from RNA polymerase II promoter (14/52), regulation of transcription, DNA-templated (21/52), RNA polymerase II core promoter proximal region sequence-specific DNA binding (9/52), transcription, DNA-templated (19/52) | 1.22e-02 |
| K | trehalose biosynthetic process (2/19), Golgi to endosome transport (3/19), ubiquitin binding (4/19) | 3.81e-02 |
| L | sequence-specific DNA binding (21/70), RNA polymerase II core promoter proximal region sequence-specific DNA binding (13/70), DNA binding (27/70), positive regulation of transcription from RNA polymerase II promoter (17/70), regulation of transcription, DNA-templated (27/70) | 6.75e-05 |
| M | structural constituent of nuclear pore (8/54), protein targeting to nuclear inner membrane (5/54), nuclear pore central transport channel (6/54), mRNA transport (9/54), nuclear pore (8/54) | 5.87e-05 |
| N | sequence-specific DNA binding (18/39), DNA binding (19/39), zinc ion binding (11/39), regulation of transcription, DNA-templated (19/39), RNA polymerase II transcription factor activity, sequence-specific DNA binding (8/39) | 6.21e-04 |
| O | regulation of transcription, DNA-templated (53/130), transcription, DNA-templated (50/130), sequence-specific DNA binding (25/130), positive regulation of transcription from RNA polymerase II promoter (26/130), nuclear-transcribed mRNA catabolic process, deadenylation-dependent decay (8/130), endocytosis decreased (26/130), invasive growth increased (37/130), cell shape abnormal (15/130) | 1.29e-02 |
| P | intracellular signal transduction (19/129), protein kinase activity (22/129), protein serine/threonine kinase activity (22/129), kinase activity (24/129), phosphorylation (24/129) | 3.34e-06 |
| Q | extracellular region (33/67), fungal-type cell wall (30/67), cell wall (25/67), anchored component of membrane (20/67), cell wall organization (23/67) | 1.01e-20 |
| R | positive regulation of transcription from RNA polymerase II promoter (21/119), DNA binding (32/119), RNA polymerase II core promoter proximal region sequence-specific DNA binding (12/119), transcription factor activity, sequence-specific DNA binding (10/119), transcription, DNA-templated (33/119) | 1.55e-02 |
| S | integral component of membrane (59/133), membrane (68/133), fungal-type vacuole membrane (18/133), vacuole (18/133), L-tyrosine transmembrane transporter activity (4/133) | 5.48e-03 |
| T | stress-activated protein kinase signaling cascade (4/33), regulation of apoptotic process (4/33) | 3.57e-02 |
| U | cytoskeleton (15/80), spindle (6/80), kinetochore microtubule (3/80) | 1.47e-02 |
| V | fungal-type vacuole (15/43), mannosylation (7/43), integral component of membrane (28/43), cell wall mannoprotein biosynthetic process (6/43), alpha-1,6-mannosyltransferase activity (4/43) | 1.45e-05 |

*Table 1 continued on next page*

*Table 1 continued*

| ID | Annotations (Positive proteins in cluster/Total proteins in cluster) | Corrected P <= |
|---|---|---|
| W | mitochondrion (144/165), mitochondrial inner membrane (57/165), mitochondrial matrix (34/165), oxidation-reduction process (31/165), mitochondrial translation (22/165), respiratory growth decreased rate (81/165), respiratory growth absent (71/165), mitochondrial genome maintenance absent (25/165) | 3.15e-15 |

The online version of this article includes the following source data for Table 1:

Source data 1. Full table of enrichments for *Table 1*.

disordered region of Emp47, which has an evolutionary signature that we predict to be associated with targeting signals for the endoplasmic reticulum (cluster V). In this case, as we predicted, we found no mitochondrial localization of Cox15-GFP. For strains where there was no discernible localization to the mitochondria (and no presence of GFP), we wanted to rule out that these were caused by the transformation process. Therefore, we restored the wildtype N-terminal IDR in the strains in which the Cox15 IDR was replaced with Emp47 or knocked out (*Figure 6—figure supplement 4*). Because we were able to rescue the mitochondrial localization with the restored wildtype IDR, we conclude that the absence of mitochondrial GFP expression is directly linked to the mutations we made in the N-terminal IDR of each strain. Importantly, these putative targeting signals have no detectable similarity when their primary amino acid sequences are aligned, and we therefore suggest that the similarity in their molecular features is preserved by stabilizing selection (see Discussion). These results confirm that IDRs with similar evolutionary signatures can rescue subcellular targeting functions, and suggest that the evolutionary signatures are specific enough to predict function of at least some IDRs.

## Evolutionary signatures of function can be used for functional annotation of fully disordered proteins

A major challenge to proteome-wide analysis of IDRs is the limited applicability of homology-based sequence analysis. Proteins with a mixture of disordered regions and structured domains can be assigned function based on homology to their structured domains, but fully disordered proteins are much more difficult to classify (reviewed in *van der Lee et al., 2014*). We therefore asked whether hypotheses about functions of fully disordered proteins could be generated using evolutionary signatures. We identified ten yeast proteins of unknown function that are predicted to be most disordered (see Materials and methods). To predict function according to our clustering analysis, we simply assigned them the annotation of the cluster in which they fell (*Table 2*). For example, Rnq1 has been extensively studied as a 'yeast prion', but there is no clear function associated with this protein under normal conditions (*Kroschwald et al., 2015*; *Sondheimer and Lindquist, 2000*; *Treusch and Lindquist, 2012*). Interestingly, Rnq1 falls into our cluster of disordered regions that are associated with nucleocytoplasmic transport (cluster M) and the nuclear pore central transport channel. While Rnq1 is annotated with a cytosolic localization, an *RNQ1* deletion was recently shown to cause nuclear aggregation of the polyQ-expanded huntingtin exon1 (Httex1) in a model of Huntington's disease (*Zheng et al., 2017*). Therefore, we propose a role for Rnq1 in nucleocytoplasmic transport. For some of these largely disordered proteins, we obtain large disordered segments falling into multiple clusters (indicated by more than one cluster ID in *Table 2*), suggesting more than one possible function for the protein (see Discussion). To estimate the predictive power of this approach, we also identified the ten most disordered yeast proteins with known functions based on their gene descriptions in SGD (*Cherry et al., 2012*). We found that predictions of function based on our cluster annotations clearly matched the gene description of the proteins in 5 of 10 cases, suggesting a predictive power of at least 50% (*Supplementary file 1* - Table S3, see Discussion). This analysis illustrates how evolutionary signatures can be used to generate hypotheses of function for fully disordered proteins.

## Discussion

In this work, we tested for evolutionary constraints on highly diverged intrinsically disordered regions proteome-wide. In contrast to the relative lack of constraint on primary amino acid sequence

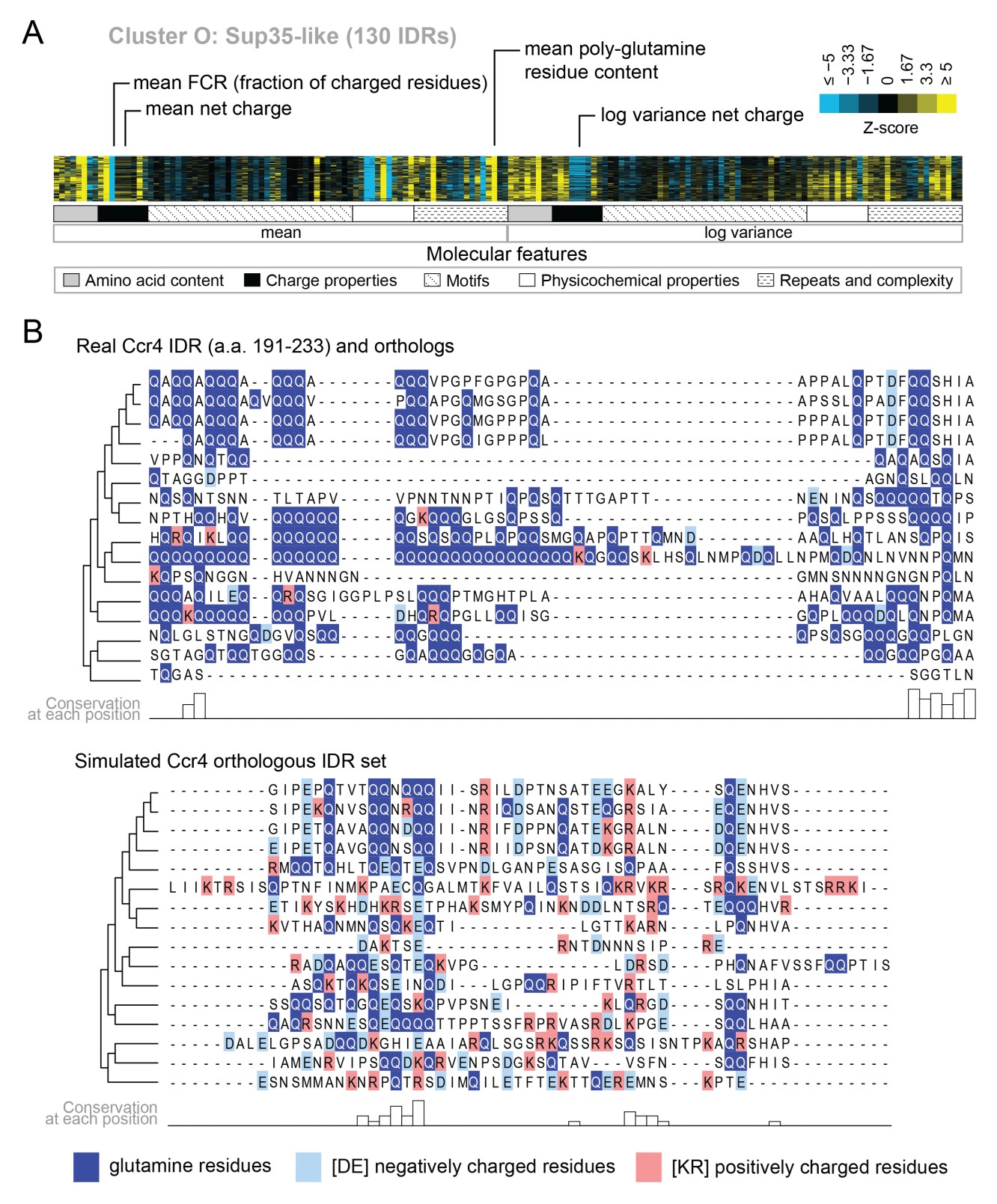

**Figure 4.** Evolutionary signatures in cluster O contain some molecular features that are typically associated with IDRs as well as some that are not. (**A**) Pattern of evolutionary signatures in cluster O. (**B**) Example disordered region from cluster O, Ccr4, with a subset of highlighted molecular features compared between its real set of orthologs and an example set of simulated orthologous IDRs. Species included in phylogeny in order from top to bottom are *S. cerevisiae*, *Saccharomyces mikatae*, *Saccharomyces kudriavzevii*, *Saccharomyces uvarum*, *Candida glabrata*, *Kazachstania naganishii*,

*Figure 4 continued*

*Naumovozyma castellii, Naumovozyma dairenensis, Tetrapisispora blattae, Tetrapisispora phaffii, Vanderwaltozyma polyspora, Zygosaccharomyces rouxii, Torulaspora delbrueckii, Kluyveromyces lactis, Eremothecium (Ashbya) cymbalariae, Lachancea waltii.*

alignments (compared to folded regions, *Brown et al., 2002*; *Tóth-Petróczy and Tawfik, 2013*), we find that the vast majority of disordered regions contain molecular features that deviate in their evolution from our null expectation (a simulation of disordered region evolution; *Nguyen Ba et al., 2014*; *Nguyen Ba et al., 2012*). Our discovery that highly diverged disordered regions contain (interpretable) molecular features that are under evolutionary constraint provides researchers with testable hypotheses about molecular features that could be important for function in their proteins of interest. Furthermore, in principle, our framework for the analysis of diverged disordered regions can be extrapolated to proteins from other species.

Importantly, our choice of features was based on previous reports of important sequence features in IDRs that could be easily calculated for protein sequences and scaled to millions of simulated sets of orthologous IDRs. Thus, our evidence for constraint must represent a lower bound on the total amount of functional constraint on highly diverged IDRs: there are very likely to be sequence characteristics that were not captured by our features. Further, even when we do find evidence for constraint on a feature, we do not know whether our feature represents the actual feature required for IDR function, or is simply correlated with it. For example, we found IDRs that show constraint on glycine and arginine content, but these may reflect the real constraint on planar-pi interactions (*Vernon and Forman-Kay, 2019*) and are not fully captured by either of these features. In the future, we could exhaustively search for protein sequence features that best explain the evolutionary patterns as was done for features of activation domains that explain reporter activity (*Ravarani et al., 2018*).

Another important element to consider is that many of our features are correlated with each other. For example, net charge is correlated with the fraction of acidic and/or basic residues in a given IDR. Due to the non-independence of molecular features that comprise the 'evolutionary signatures', it is difficult to quantify similarity between these evolutionary signatures using standard distance metrics that assume independence. We use a weighted uncentred correlation distance (*de Hoon et al., 2004*) in the clustering analysis, which limits the impact of correlated features. However, we caution against a quantitative comparison of individual evolutionary signatures.

Despite our choice of molecular features, we found strong evidence that groups of disordered regions share evolutionary signatures, and that these groups of IDRs are associated with specific biological functions. To demonstrate the association of evolutionary signatures with previously known functions, we associated IDRs with protein function, and indeed showed that we could generate hypotheses about protein function for fully disordered proteins. However, many proteins contain multiple IDRs. In these proteins, the IDRs may perform different functions (just as multiple folded domains may perform independent functions), thus complicating the mapping of molecular functions to molecular features of IDRs. We are currently developing more well-defined statistical approaches to predict function for IDRs based on evolutionary signatures. Systematic data at the level of individual IDRs would greatly facilitate future progress in this area.

Another challenge in associating specific functions with individual IDRs is that current bioinformatics predictions of IDRs at the proteome level often lead to arbitrary breaks (or merging) of IDRs, as IDR boundaries are very difficult to define precisely (even with sensitive experimental approaches; *Jensen et al., 2013*). Whether or not IDRs serve as distinct functional units across a linear peptide sequence, and where the boundaries for these regions lie on a proteome-wide scale, is an area for further research. In our cluster analysis, we find that the vast majority of IDRs in multi-IDR proteins fall into different clusters, and that this matches our expectation from random chance. A small minority of IDRs from very large (>1500 amino acid) disordered proteins cluster together, suggesting that they are 'broken up' pieces of larger units.

Despite the caveat of IDR boundaries in proteome-wide analyses, evolutionary signatures of selection on molecular features represent a new way to assign function to the large numbers of currently enigmatic IDRs that have been identified based on protein sequences. This approach is complementary to current bioinformatics approaches to predict IDR function that are based on presence (*Edwards et al., 2007*) or conservation of SLiMs (*Beltrao and Serrano, 2005*; *Davey et al., 2012*;

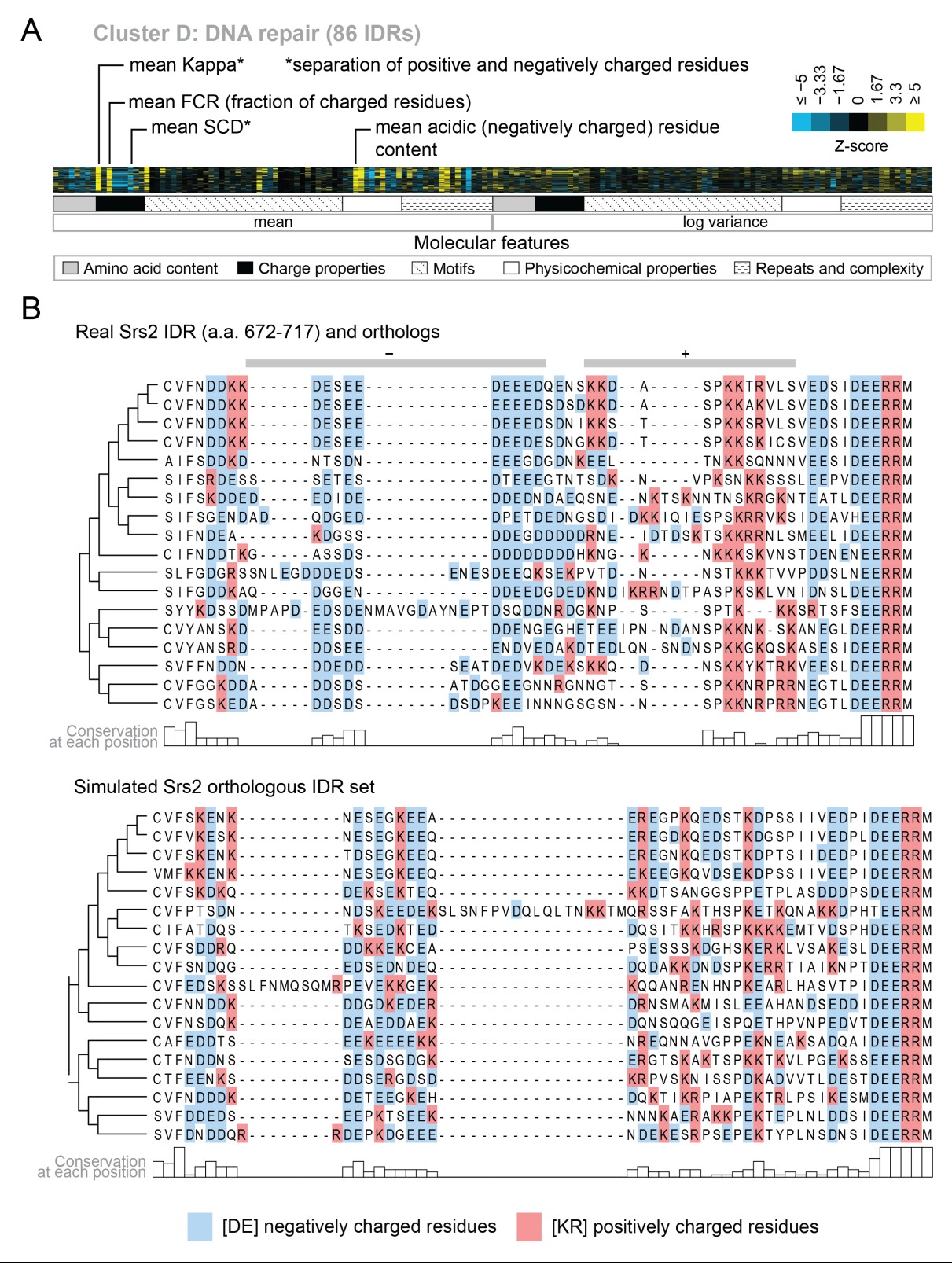

**Figure 5.** Cluster D contains disordered regions associated with DNA repair. (**A**) Pattern of evolutionary signatures in cluster D. (**B**) Example disordered region from cluster D, Srs2, with a subset of highlighted molecular features compared between its real set of orthologs and an example set of simulated orthologous IDRs. Species included in phylogeny in order from top to bottom are *S. cerevisiae*, *S. mikatae*, *S. kudriavzevii*, *S. uvarum*,

*Figure 5 continued on next page*

*Figure 5 continued*

*C. glabrata, Kazachstania africana, K. naganishii, N. castellii, N. dairenensis, T. phaffii, Z. rouxii, T. delbrueckii, K. lactis, Eremothecium (Ashbya) gossypii, E. cymbalariae, Lachancea kluyveri, Lachancea thermotolerans, L. waltii.*

*Lai et al., 2012*; *Nguyen Ba et al., 2012*), prediction of interactions (MoRFs) (*Fuxreiter et al., 2004*; *Lee et al., 2012*; *Mohan et al., 2006*; *Oldfield et al., 2005*; *Vacic et al., 2007*), or prediction of phase separation (*Vernon and Forman-Kay, 2019*).

Our evolutionary analysis addresses challenges in assigning function to highly diverged intrinsically disordered regions whose presence is conserved in orthologs. Although there are examples of slowly evolving disordered regions (*Ahrens et al., 2016*; *Ahrens et al., 2018*; *Brown et al., 2002*), these reflect a minority of IDRs, and pose less of a challenge for standard protein sequence analysis methods. Because we take the evolutionary rate of IDRs into account in our analysis, we do not identify constrained molecular features for these slowly evolving IDRs, as they would not be significantly different in our simulated IDRs. Furthermore, although there are examples of IDRs that are lost or gained through evolution (*Ahrens et al., 2017*; *Montanari et al., 2011*), the presence of IDRs is conserved across orthologs in the vast majority of cases (*Bellay et al., 2011*; *Chen et al., 2006a*; *Chen et al., 2006b*; *Colak et al., 2013*).

Widespread evidence for shared functions in the highly diverged portions of IDRs also has several evolutionary implications. The lack of homology between most IDRs with similar evolutionary signatures suggests that the molecular features are preserved in each IDR independently. For example, the more than 150 IDRs that we believe represent mitochondrial N-terminal targeting signals share similar constraints on their molecular features, yet these signals have been preserved independently over very long evolutionary time as mitochondrial genes were transferred individually to the nuclear genome (*Adams and Palmer, 2003*). The preservation of molecular features over long evolutionary time, despite accumulation of amino acid divergence, is consistent with a model of stabilizing selection (*Bedford and Hartl, 2009*; *Hansen, 1997*; *Lande, 1976*), where individual amino acid sites are under relatively weak functional constraints (*Landry et al., 2014*). In this view, single point mutations are unlikely to dramatically impair IDR function, and therefore large evolutionary divergence can accumulate. This also suggests that disease-causing mutations in disordered regions are more likely to cause gain of function, consistent with at least one recent study (*Meyer et al., 2018*).

Although current models for the evolution of short linear motifs (well-characterized functional elements in IDRs) also implicate stabilizing selection (*Koch et al., 2018*; *Landry et al., 2014*), these motifs represent only a minority of the residues in disordered regions (*Nguyen Ba et al., 2012*). Our observation of shared evolutionary signatures associated with specific functions in highly diverged IDRs suggests that this evolutionary mechanism is shaping the proteome on a much wider scale than currently appreciated. Further, stabilizing selection stands in contrast to purifying selection on individual residues, the major evolutionary mechanism thought to preserve function in stably folded regions of the proteome (*Taylor and Raes, 2004*). Thus, we propose that these two major biophysical classes of protein regions (IDRs vs. folded regions) also evolve under two different functional regimes.

## Materials and methods

### Multiple sequence alignments and visualization

We acquired orthologs of *Saccharomyces cerevisiae* from the Yeast Gene Order Browser (*Byrne and Wolfe, 2005*) and made multiple sequence alignments using MAFFT (*Katoh and Standley, 2013*) with default settings, as previously described (*Nguyen Ba et al., 2014*; *Nguyen Ba et al., 2012*). We visualized multiple sequence alignments using Jalview (*Waterhouse et al., 2009*).

### Quantification of evolutionary divergence of IDRs and ordered regions of the proteome

We identified IDRs in the *S. cerevisiae* proteome using DISOPRED3 (*Jones and Cozzetto, 2015*) and filtered them to include only those that are 30 amino acids or longer. We identified the non-disordered regions of the proteome as the inverse subset of the IDRs, and again only included regions

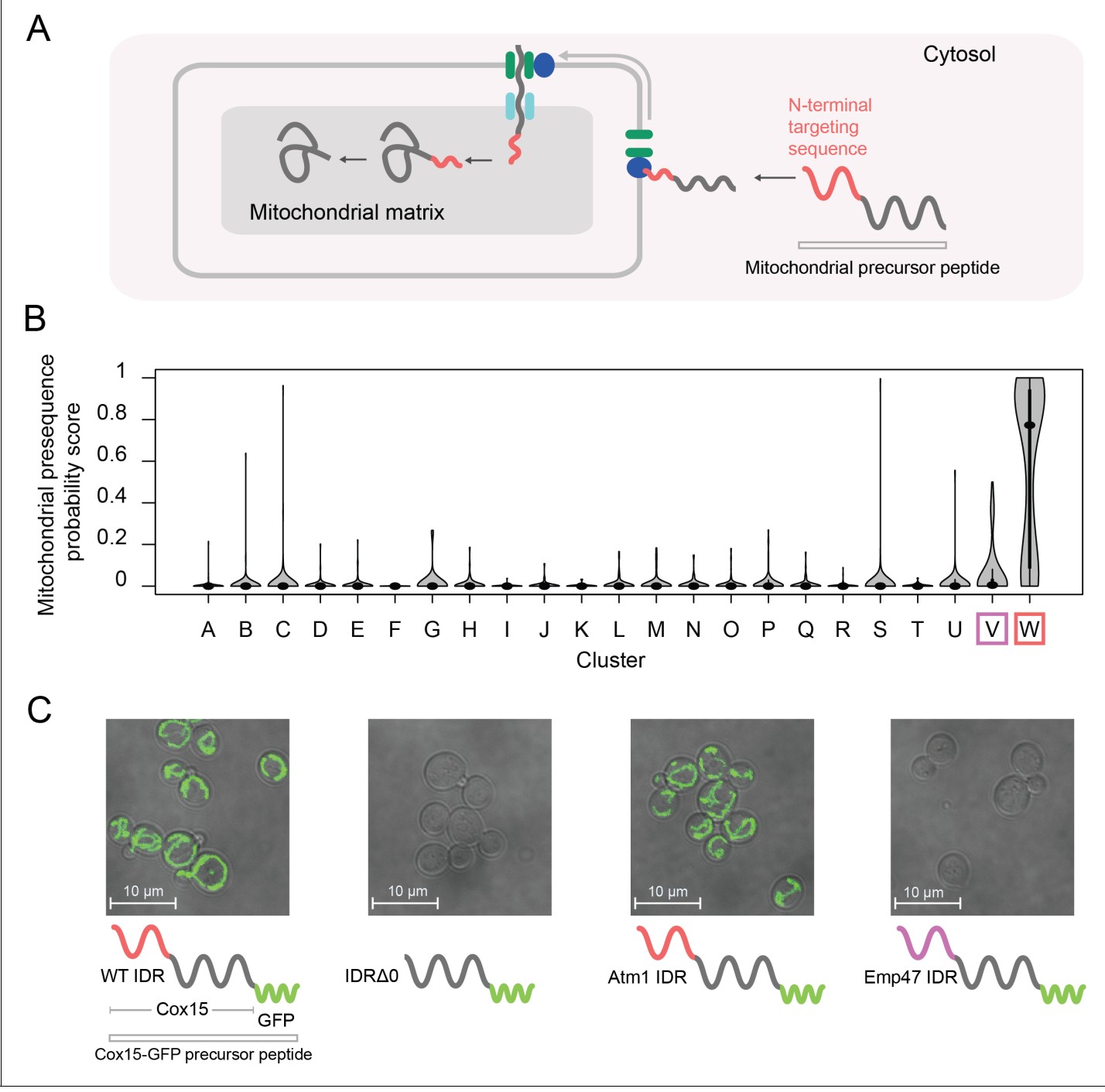

**Figure 6.** Cluster W is associated with mitochondrial N-terminal targeting signals. (**A**) Schematic (not to scale) showing the path of a mitochondrial precursor peptide (with N-terminal targeting sequence in red) from the cytosol, where it is translated, to the mitochondrial matrix, where the peptide folds and targeting sequence is cleaved. (**B**) Violin plots (median indicated by black dot, thick black line showing 25th-75th percentile, and whiskers showing outliers) show distributions of mitochondrial presequence probability scores for all IDRs in each cluster. The cluster that we predict to contain mitochondrial N-terminal targeting signals is outlined in red, while the cluster that we predict to contain endoplasmic reticulum targeting signals is outlined in purple. (**C**) Micrographs of *S. cerevisiae* strains in which Cox15 is tagged with GFP, with either the wildtype Cox15 IDR, deletion of the Cox15 IDR, replacement of the Cox15 IDR with the Atm1 IDR (also in the mitochondrial targeting signal cluster), or replacement of the Cox15 IDR with the Emp47 IDR (from the endoplasmic reticulum targeting signal cluster).

The online version of this article includes the following figure supplement(s) for figure 6:

*Figure 6 continued on next page*

*Figure 6 continued*

**Figure supplement 1.** Evolutionary signatures in cluster W contain molecular features that have been previously reported for mitochondrial N-terminal targeting signals.

**Figure supplement 2.** Full field-of-view micrographs of *S. cerevisiae* strains from *Figure 6C*.

**Figure supplement 3.** Micrographs of *S. cerevisiae* strains with three different genotypes.

**Figure supplement 4.** Reverse transformation of GFP-tagged Cox15 IDRΔ0 and Cox15ΔEmp47 strains to wildtype Cox15 IDR rescues mitochondrial localization of Cox15-GFP.

that are 30 amino acids or longer. Using the multiple sequence alignments constructed for these protein regions (as above), and only including those proteins for which there at least 10 species in the alignment and at least 10 amino acids for each species, we calculated evolutionary distances for each region using PAML (*Yang, 2007*) using the WAG model, with an initial kappa of 2, initial omega of 0.4, and clean data set to 0. We used the sum of branch lengths for each region to estimate the evolutionary divergence, and plotted the distribution of this metric for IDRs and non-IDRs in the *S. cerevisiae* proteome in *Figure 1—figure supplement 1*.

## Quantification of IDR overlap with pfam annotations

We obtained the list of Pfam (*El-Gebali et al., 2019*) domain coordinates for *S. cerevisiae* from the Saccharomyces Genome Database (SGD) (*Cherry et al., 2012*). We included domain coordinates that had e-values less than or equal to 1, and which occurred in more than one protein in the *S. cerevisiae* proteome. We then computed the percentage overlap of each IDR (coordinates determined as above) with the Pfam domain coordinates, and plotted the distribution of percent overlap values for all predicted IDRs in the *S. cerevisiae* proteome in *Figure 1—figure supplement 2*.

## Evolutionary analysis of diverged disordered regions

Evolutionary analysis of diverged disordered regions was performed as in *Zarin et al. (2017)*, with some modifications to facilitate proteome-wide analysis. Briefly, we sought to simulate evolution of disordered regions while being agnostic to the conservation of molecular features that we tested. Using the multiple sequence alignments of *S. cerevisiae* IDRs and species branch lengths (as described above), we used the previously described phylo-HMM software (*Nguyen Ba et al., 2014*; *Nguyen Ba et al., 2012*) to obtain maximum likelihood estimates of the site-specific (i.e. 'column') and 'local' rate of evolution for each amino acid. The phylo-HMM exploits preferential conservation of SLiMs in disordered regions to predict them by comparing these site-specific rates to the 'local'

**Table 2.** Evolutionary signatures of function can be used for functional annotation of previously uncharacterized proteins and IDRs.

| ID | Name | Description | % Disorder | Cluster ID |
|---|---|---|---|---|
| YCL028W | RNQ1 | Protein whose biological role is unknown; localizes to the cytosol | 96 | M: Nucleocytoplasmic transport |
| YKL105C | SEG2 | Protein whose biological role is unknown; localizes to the cell periphery | 92 | P: Signal transduction |
| YGR196C | FYV8 | Protein whose biological role is unknown; localizes to the cytoplasm in a large-scale study | 89 | A: Ribosome biogenesis R: Transcription |
| YGL023C | PIB2 | Protein whose biological role is unknown; localizes to the mitochondrion in a large-scale study | 86 | R: Transcription |
| YOL036W | | Protein whose biological role and cellular location are unknown | 84 | P: Signal transduction R: Transcription |
| YNL176C | TDA7 | Protein whose biological role is unknown; localizes to the vacuole | 83 | Q: Cell wall organization |
| YFR016C | | Protein whose biological role is unknown; localizes to both the cytoplasm and bud in a large-scale study | 83 | A: Ribosome biogenesis |
| YBL081W | | Protein whose biological role and cellular location are unknown | 82 | M: Nucleocytoplasmic transport |
| YBR016W | | Protein whose biological role is unknown; localizes to the bud membrane and the mating projection membrane | 82 | O: Sup35-like |
| YOL070C | NBA1 | Protein whose biological role is unknown; localizes to the bud neck and cytoplasm and colocalizes with ribosomes in multiple large-scale studies | 81 | Does not fall into annotated cluster; close to ribosome biogenesis cluster |

rates of evolution. Because indels are frequently found in disordered regions, the phylo-HMM software models sequence evolution as a probabilistic evolutionary process that considers amino acid substitutions according to a standard continuous time markov process framework (*Durbin et al., 1998*) and generation of indels with power-law distribution (*Cartwright, 2006*). For our set of data, we used the default parameters of the software, that is the parameters used in the original study on a similar set of data (for example, we used a window size of 31 alignment columns to estimate the 'local' rate of evolution).

Having predicted these rates of evolution and conserved SLiMs for each IDR, we simulated 1000 orthologous sets of IDRs for each IDR. We did this using a previously described disordered region evolution simulator (*Nguyen Ba et al., 2014*; *Nguyen Ba et al., 2012*) that preserves SLiMs and evolves disordered regions according to a disordered region-specific substitution matrix, but is agnostic to other molecular features. We used the *S. cerevisiae* sequence as the root for the simulator. The simulator uses a model similar to the phylo-HMM where amino acids can either mutate or generate an indel according to the specified rates of evolution. In contrast to the phylo-HMM, the substitution matrix used in the simulator was obtained using the stationary frequencies of the 20 amino acids derived from predicted disordered regions in the *S. cerevisiae* proteome, and exchangeability between pairs of amino acids was inferred from closely related species *S. cerevisiae*, *Saccharomyces paradoxus*, *S. mikatae*, and *Saccharomyces bayanus* (as in *Nguyen Ba et al., 2014*; *Nguyen Ba et al., 2012*). Therefore, as evolutionary time increases, sequences diverge from the initial *S. cerevisiae* sequence according to the phylogenetic tree and tend towards the equilibrium stationary frequencies of disordered amino acids. This simulator requires a scaling factor to convert evolutionary distances from substitutions per site as obtained from PAML (*Yang, 2007*). We chose the scaling factor such that the average distance between *S. cerevisiae* and *S. uvarum* over all the IDR alignments equals 1.

Sequences and trees were read into R using the 'seqinr' (*Charif and Lobry, 2007*) and 'ape' (*Paradis and Schliep, 2019*) packages, respectively. Sequences were parsed in R using the 'stringr' (*Wickham, 2010*) and 'stringi' (*Gagolewski, 2019*) packages. We calculated all the sequence features for the real and simulated set of IDR orthologs using custom functions in R except for isoelectric point, which was computed using the computePI function in seqinr (*Charif and Lobry, 2007*), as well as 'Omega' (*Martin et al., 2016*), 'Kappa' (*Das and Pappu, 2013*), and Wootton-Federhen complexity (*Wootton and Federhen, 1993*), which were calculated using the localCider program (*Holehouse et al., 2017*) called through R using the 'rPython' package (*Bellosta, 2015*).

We calculated the mean and log variance of each feature for each real set of orthologous IDRs and each of the 1000 sets of orthologous IDRs. Because simulations sometimes lead to the deletion of the IDR, we did not include those IDRs that had fewer than 950 non-empty simulations. To obtain a random expectation for *Figure 1C*, we quantified the number of significant ($p<0.01$) molecular features in a set of randomly chosen simulated IDRs (one for each real IDR).

To summarize the difference between each real set of orthologous IDRs and its corresponding 1000 simulated sets of orthologous IDRs, we used a standard Z-score ($Z$) where we subtracted the mean of the simulations ($\mu$) from the real value ($x$) and divided by the standard deviation of the simulations ($\sigma$). The formula for the Z-score is as follows:

$$Z = \frac{x - \mu}{\sigma}$$

## Strain construction and growth conditions

All strains (*Supplementary file 1* - Table S4) were constructed in the *S. cerevisiae* BY4741 background. IDR transformants were constructed using the *Delitto Perfetto* in vivo site-directed mutagenesis method (*Storici et al., 2001*). Ste50 IDR mutants were constructed in the ssk22Δ0::HisMx3 ssk2Δ0 background as in *Zarin et al. (2017)*. Genomic changes in transformed strains were confirmed by Sanger sequencing. For mitochondrial strains, starting strains were acquired from the GFP collection (*Huh et al., 2003*). The Fus1pr-GFP reporter was constructed as in *Zarin et al. (2017)* using Gibson assembly (*Gibson et al., 2009*), integrated at the *HO* locus using a selectable marker (URA3), and confirmed by PCR.

All experiments were done on log-phase cells grown at 30°C in rich or synthetic complete media lacking appropriate nutrients to maintain selection of markers, unless otherwise stated. Two percent (wt/vol) glucose was used as the carbon source.

## Confocal microscopy and image analysis

We acquired all images with a Leica TCS SP8 microscope using standard, uncoated glass slides with a 100x objective. For all GFP images, seven evenly spaced z-slices covering ~6 microns in the z plane were collected for each field of view, and maximum projections of these slices were quantified for *Figure 2B*, or presented as micrographs in *Figure 6C*. To quantify basal Fus1pr-GFP expression, single cells in micrographs were segmented using YeastSpotter (http://yeastspotter.csb.utoronto.ca/) (*Lu et al., 2019*). The segmented masks and corresponding fluorescent images were imported into R using the 'EBImage' package (*Pau et al., 2010*), and GFP intensity for each cell was quantified using base R functions (example available on http://yeastspotter.csb.utoronto.ca). To assay shmooing, log phase cells were inoculated with 1 uM alpha factor for 2 hr at 30˚C (as in *Kompella et al., 2016*), at which point they were imaged in brightfield as above. We repeated each microscopy experiment at least twice on different days, and present representative results from one of those days in *Figure 2B, C* and *Figure 6*.

## Clustering of proteome-wide evolutionary signatures

Hierarchical clustering was performed using the Cluster 3.0 program (*de Hoon et al., 2004*). The evolutionary signature data was first filtered to include only those IDRs that had at least one Z-score with an absolute value of 3 or more, and with at least 95% data present for the 164 features. This resulted in 4646 IDRs (filtered from the initial 5149) that were then clustered using uncentered correlation distance and average linkage, with 'cluster' and 'calculate weights' options selected for 'genes' (i.e. IDRs), but not for arrays (i.e. molecular features). Clusters were picked manually for further analysis. The full clusterplot is available in supplementary data.

In order to ensure that the clustering was not simply due to sequence similarity between the disordered regions, for each cluster, we compared the pairwise distance of the IDRs in each cluster to that of the IDRs outside that cluster, and calculated the percent of disordered regions in each cluster that fell in the top 1% of pairwise distance in all the clusters. This metric is presented for each cluster in *Supplementary file 1* - Table S2. For example, the cluster with the highest amount of sequence similarity according to this threshold (top 1% similarity) is cluster Q, with 8.9% of IDRs in this cluster having high sequence similarity according to this threshold. However, the vast majority of the clusters are comprised of IDRs with negligible sequence similarity; for example, 17/23 clusters have less than 1% IDRs with high sequence similarity. In order to calculate pairwise distances, we made pairwise global alignments (with gap opening penalty of 0 and gap extension penalty of 1) and calculated distances using the BLOSUM62 matrix (this was done using the 'stringDist' function in the Biostrings R package; *Pagès et al., 2018*).

## Tests for enrichment of annotations

Annotations for Gene Ontology (GO) terms, phenotypes, and literature were acquired from SGD (*Cherry et al., 2012*) for the *S. cerevisiae* proteome. We included GO terms that applied to a maximum of 5000 genes in the *S. cerevisiae* proteome. A test for enrichment of annotations was done using the hypergeometric test for each cluster against all the proteins in the clustering analysis. To obtain Q-values, p-values were corrected using the Benjamini-Hochberg method. Q-values below an FDR of 5% were retained. Because there is not a 1-to-1 correspondence between IDRs and annotations, which are based on proteins, we also calculated Q-values using permutation tests. To do so, we uniformly sampled 1000 clusters of IDRs for each cluster from the 4646 IDRs included in our clusterplot, and obtained the sum of the top ten – log Q-values associated with each test for enrichment, as above. We compared this test statistic to the observed sum of top ten – log Q-values for each cluster, and reported the difference as a standard Z-score in *Supplementary file 1* - Table S2.

In order to understand how our evolutionary signatures compare to information obtained only from amino acid frequencies, we computed vectors of Z-scores for each IDR that represented their amino acid frequencies normalized to the proteome-wide average. We clustered these vectors using k-means (K = 25) with the Cluster 3.0 program (*de Hoon et al., 2004*) We performed a similar permutation test (as above), where the sample of 1000 clusters was not uniform, but drawn to create 1000 random clusters of IDRs with similar amino acid composition for each cluster. For example, for each IDR in a cluster, we found the cluster that it fell into in the amino acid frequency clusterplot, and sampled from that cluster to replace the IDR in our evolutionary signature clusterplot. We did

this 1000 times for each cluster, and used the same test statistic as the above-described permutations to report the difference in enriched annotations between our clusterplot based on evolutionary signatures and the clusterplot based on amino acid frequencies (*Supplementary file 1* - Table S2).

### Identification of highly disordered proteins with unknown function

We identified proteins whose biological role is unknown according to their SGD annotation (*Cherry et al., 2012*). We quantified the percent of residues that were predicted to be disordered in each protein with unknown function, and present the top ten most disordered proteins in *Table 2*.

## Acknowledgements

We thank Alex X Lu, Dr. Christiane Iserman, Dr. Iva Pritišanac, Shadi Zabad, and Ian S Hsu for comments on the manuscript. We thank Alex X Lu for stimulating discussions about clustering and Dr. Iva Pritisanac for suggesting analysis of completely disordered proteins. We thank Dr. Helena Friesen and Dr. Brenda Andrews for providing strains from the yeast GFP collection. We thank Canadian Institutes for Health Research (CIHR) for funding to AMM and JDF-K (grant no. PJT-148532), the Canada Research Chairs program and a CIHR Foundation grant (grant no. FDN-148375) to JDF-K, Canada Foundation for Innovation (CFI) for funding to AMM, and the Natural Sciences and Engineering Research Council of Canada (NSERC) for an Alexander Graham Bell scholarship and Michael Smith Foreign Study Supplement to TZ.

## Additional information

### Funding

| Funder | Grant reference number | Author |
|---|---|---|
| Natural Sciences and Engineering Research Council of Canada | Alexander Graham Bell Scholarship | Taraneh Zarin |
| Natural Sciences and Engineering Research Council of Canada | Discovery Grant | Alan M Moses |
| Canadian Institutes of Health Research | PJT-148532 | Julie D Forman-Kay Alan M Moses |
| Canadian Institutes of Health Research | FDN-148375 | Julie D Forman-Kay |
| Canada Research Chairs | | Julie D Forman-Kay |
| Canada Foundation for Innovation | | Alan M Moses |
| Natural Sciences and Engineering Research Council of Canada | Postdoctoral Fellowship | Alex N Nguyen Ba |

The funders had no role in study design, data collection and interpretation, or the decision to submit the work for publication.

### Author contributions

Taraneh Zarin, Conceptualization, Data curation, Software, Formal analysis, Funding acquisition, Validation, Investigation, Visualization, Methodology, Writing—original draft, Project administration, Writing—review and editing; Bob Strome, Validation; Alex N Nguyen Ba, Data curation, Software, Writing—review and editing; Simon Alberti, Resources, Supervision; Julie D Forman-Kay, Conceptualization, Resources, Funding acquisition, Writing—review and editing; Alan M Moses, Conceptualization, Resources, Software, Formal analysis, Supervision, Funding acquisition, Investigation, Methodology, Writing—original draft, Project administration, Writing—review and editing

## Author ORCIDs
Taraneh Zarin (ID) https://orcid.org/0000-0003-1253-3843
Simon Alberti (ID) http://orcid.org/0000-0003-4017-6505
Julie D Forman-Kay (ID) https://orcid.org/0000-0001-8265-972X
Alan M Moses (ID) https://orcid.org/0000-0003-3118-3121

## Decision letter and Author response
Decision letter https://doi.org/10.7554/eLife.46883.sa1
Author response https://doi.org/10.7554/eLife.46883.sa2

## Additional files

### Supplementary files
• Supplementary file 1. Includes Tables S1-S4.

• Transparent reporting form

### Data availability
The analysis is based on publicly available sequence data from the Yeast Gene Order Browser (YGOB). Source data has been included as supplementary data.

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
