## [Decision Letter]

Thank you for submitting your article "Proteome-wide signatures of function in highly diverged intrinsically disordered regions" for consideration by *eLife*. Your article has been reviewed by Michael Eisen as the Senior Editor and Reviewing Editor, and two reviewers. The following individuals involved in review of your submission have agreed to reveal their identity: Xavier Darzacq (Reviewer #1) and Jessica Siltberg-Liberles (Reviewer #2).

The reviewers have discussed the reviews with one another and the Reviewing Editor has drafted this decision to help you prepare a revised submission. Please aim to submit the revised version within two months.

Summary:

While related protein intrinsically disordered regions (IDRs) usually have little sequence similarity, a growing body of work suggests that general molecular features of these sequences – such as charge or proportions of specific amino acid residues – are conserved. In this manuscript, Zarin et al.et al., find evidence for such conservation by extending a methodology previously applied to an IDR from the *S. cerevisiae* protein Ste50p to a broader collection of IDRs from the *S. cerevisiae* proteome. The reviewers consider the manuscript to be a well-executed and timely stud y of interest to a wide range of *eLife* readers, and suggest only a series of relatively minor revisions.

Essential revisions:

The only major point raised by the reviewers is that the manuscript would be improved with a more in depth description of the methods, which are referred to currently primarily by citation.

For example, it is unclear whether the same power law distribution for gap lengths is used as in past work by Nguyen Ba et al.et al., and how the disordered region substitution matrices are inferred. Additionally, a brief summary of the phylo-HMM(s) (both sequence and gap) used to infer sequence conservation in individual IDRs would be useful given how central this is to the results in the manuscript. Such a discussion could be accomplished simply by extending the existing subsection "Evolutionary analysis of diverged disordered regions".

Minor points:

1) In Figure 6C and Figure 6—figure supplement 2, no fluorescence is visible for Cox15 IDR∆0 and Cox15∆Emp47. This makes it hard to judge whether these constructs fail to localize to mitochondria, as claimed in the manuscript. Are these constructs poorly expressed, or is high thresholding in the fluorescence channel responsible?

2) In Figure 2, three substitutions in the Ste50p IDR are made to demonstrate that similarity in the set of molecular features under selection ("evolutionary signature") implies the ability to perform similar functions. We recommend that the authors indicate how many consensus MAPK phosphorylation sites are present in each of these substituted domains, as this is a core function of the Ste50p IDR.

3) In Figure 2, IDRs are ordered by the Euclidean distance of their vectors of molecular feature Z-scores. This is would be appropriate for linearly independent, orthogonal vector components. Because many of the molecular features are linearly dependent (for instance, (fraction of basic residues) + (fraction of acidic residues) = (fraction of charged residues)), their Z-scores might be suspected to lack independence as well. No evidence to the contrary is provided in the manuscript. Codependence of molecular features could also bias the clustering algorithm so that it is more dependent on features that have a high redundancy (e.g. any of the charge attributes). Zarin et al., use clustering in a qualitative way, so we think this is a minor concern and should not impact the main claims of the paper. However, since this method is likely to be implemented by others, we recommend including a note of caution about using Euclidean distance to interpret differences in evolutionary signatures, especially since Figure 2 implies that this is valid.

4) The authors assert that their IDR cluster-function assignments can be used for functional annotation of 10 previously uncharacterized disordered proteins (Table 2). While this would be useful, the authors provide little validation for these predictions apart from some indirect evidence for one of the proteins (Rnq1p). To make this point more persuasive and to demonstrate the accuracy of the cluster-function assignments, the authors could include predictions for the function of 10 fully disordered proteins of known function, in addition to the existing set of proteins of unknown function.

5) The authors refer to the molecular features as "evolutionary signatures" but wouldn't "functional signatures" be a better term? The title also hints at this…

6) The authors often use the term homology when they mean sequence similarity. They are not equivalent. For example, from subsection “Clustering of proteome-wide evolutionary signatures”:

"For example, the cluster with the highest amount of "homologous" IDRs according to this threshold (top 1% homology) is cluster Q, with 8.9% homologous IDRs. However, the vast majority of the clusters have negligibly homologous IDRs; for example, 17/23 clusters have less than 1% homology between IDRs."

It would be better to write:

"For example, the cluster with the most similar IDRs according to this threshold (top 1% similarity) is cluster Q, with 8.9% sequence identity across the IDRs. However, the vast majority of the clusters have less similar IDRs; for example, 17/23 clusters have less than 1% sequence identity between IDRs."

7) Following up on the entire paragraph from subsection “Clustering of proteome-wide evolutionary signatures” mentioned above, how are the sequences from different IDRs in the clusters compared. It says with Blosum62, but how are they aligned? Gap penalties? What information do pairwise distances between sequences that are this divergent provide?

8) Disorder is predicted using DISOPRED3 on the S.cerevisiae proteome, but if disorder is only predicted for one species, how do you know that disorder is conserved for sequences in each alignment? Disorder is not necessarily conserved across sequences, not even across orthologs, see Montanari et al., 2011 and for further discussion see Ahrens et al.2017.

9) It is stated on numerous occasions that disordered regions are rapidly evolving and they most frequently do, but not always. Some disordered regions that also are predicted to have secondary structure to evolve slow, see Ahrens et al., 2018 and Ahrens et al., 2016.

10) Another recent paper that discusses functional constraint in disordered regions and due to its relevance, it ought to be referred to from the current manuscript, see Afanasyeva et al., 2018.

11) Figure 1—supplementary figure 2 shows nothing. It is supposed to show very little, but really, this shows nothing. If this is correct, can you add what the data looks like for ordered regions as a comparison?

---

## [Author Response]

Essential revisions:The only major point raised by the reviewers is that the manuscript would be improved with a more in depth description of the methods, which are referred to currently primarily by citation.For example, it is unclear whether the same power law distribution for gap lengths is used as in past work by Nguyen Ba et al.et al., and how the disordered region substitution matrices are inferred. Additionally, a brief summary of the phylo-HMM(s) (both sequence and gap) used to infer sequence conservation in individual IDRs would be useful given how central this is to the results in the manuscript. Such a discussion could be accomplished simply by extending the existing subsection "Evolutionary analysis of diverged disordered regions".

We thank the reviewers for this comment. We have now expanded the subsection “Evolutionary analysis of diverged disordered regions” to include information about the specific parameters used to infer sequence conservation via phylo-HMMs, as well as information about how the disordered region substitution matrices are inferred.

Minor points:1) In Figure 6C and Figure 6—figure supplement 2, no fluorescence is visible for Cox15 IDR∆0 and Cox15∆Emp47. This makes it hard to judge whether these constructs fail to localize to mitochondria, as claimed in the manuscript. Are these constructs poorly expressed, or is high thresholding in the fluorescence channel responsible?

In Figures 6C and Figure 6—figure supplement 2, we are making the deletion and replacement of the IDR in the genome, and we confirm these genomic changes through PCR and sequencing, as described in the Methods.

Given the microscope images, we have no evidence that these peptides are being expressed and folded in the mitochondria or elsewhere in the cell. We are using the same threshold and laser power across the images that we compare. Lowering the threshold for the Cox15 IDR∆0 and Cox15 IDR∆Emp47 strains only shows autofluorescence.

While we think it is unlikely that the transformation of these strains caused the degradation/mislocalization of peptides (particularly since we do see mitochondrial localization in the Cox15 IDR∆Atm1 strain), we nevertheless performed an experiment to rule out this possibility by re-transforming the Cox15 IDR∆0 and Cox15 IDR∆Emp47 strains to restore the wildtype N-terminal IDR. Indeed, when we restore the IDR of these strains to that of the wildtype, we restore mitochondrial localization, suggesting that it is the specific mutation in the N-terminal IDR that determines the localization pattern we see in the microscopy images (Figure 6—figure supplement 4). We reported these results in subsection “A cluster of evolutionary signatures is associated with N-terminal mitochondrial targeting signals”

Our understanding of the mitochondrial targeting system is that there are several control mechanisms that ensure peptides are degraded if they do not localize correctly (Weidberg et al., 2018; Mårtensson et al., 2019). Thus, we believe that the lack of mitochondrial (and any other) GFP signal in the Cox15 IDR∆0 and Cox15∆Emp47 strains, is consistent with the model that the mutant peptides have been degraded and/or have not reached the mitochondria. We have now provided this contextual information in the text in subsection “A cluster of evolutionary signatures is associated with N-terminal mitochondrial targeting signals”.

2) In Figure 2, three substitutions in the Ste50p IDR are made to demonstrate that similarity in the set of molecular features under selection ("evolutionary signature") implies the ability to perform similar functions. We recommend that the authors indicate how many consensus MAPK phosphorylation sites are present in each of these substituted domains, as this is a core function of the Ste50p IDR.

We have now included the information about the number of consensus phosphorylation sites for the Ste50 IDR as well as the substituted IDRs (Pex5, Stp4, and Rad26) in the legend for Figure 2. We find no correlation between the number of consensus phosphorylation sites and the capacity to restore function, which is consistent with our previous work (Zarin et al., 2017)

3) In Figure 2, IDRs are ordered by the Euclidean distance of their vectors of molecular feature Z-scores. This is would be appropriate for linearly independent, orthogonal vector components. Because many of the molecular features are linearly dependent (for instance, (fraction of basic residues) + (fraction of acidic residues) = (fraction of charged residues)), their Z-scores might be suspected to lack independence as well. No evidence to the contrary is provided in the manuscript. Codependence of molecular features could also bias the clustering algorithm so that it is more dependent on features that have a high redundancy (e.g. any of the charge attributes). Zarin et al., use clustering in a qualitative way, so we think this is a minor concern and should not impact the main claims of the paper. However, since this method is likely to be implemented by others, we recommend including a note of caution about using Euclidean distance to interpret differences in evolutionary signatures, especially since Figure 2 implies that this is valid.

We agree with the reviewers that the features in the vector of Z-scores should not be interpreted independently. We do not believe that this is a problem limited to Euclidian distance, specifically, but any distance that treats features independently. However, although the features are not independent, we limit the impact of co-dependence of molecular features on the clustering algorithm by using a weighted uncentred correlation distance (also known as weighted Eisen cosine distance) in the clustering analysis. These weights are chosen to downweight correlated features. As the reviewers point out, we also interpret the clustering qualitatively.

Correlated features make interpretation in high-dimensional space challenging. In Figure 2, we used Euclidian distance because it agrees with our visual interpretation of how similar the Z-score vectors are to each other. However, as the reviewers suggested, we have now added a note of caution (to the Discussion section, with referral from the Results section) that distance between vectors of Z-scores should not be used to assess similarity between evolutionary signatures of IDRs.

4) The authors assert that their IDR cluster-function assignments can be used for functional annotation of 10 previously uncharacterized disordered proteins (Table 2). While this would be useful, the authors provide little validation for these predictions apart from some indirect evidence for one of the proteins (Rnq1p). To make this point more persuasive and to demonstrate the accuracy of the cluster-function assignments, the authors could include predictions for the function of 10 fully disordered proteins of known function, in addition to the existing set of proteins of unknown function.

We thank the reviewers for suggesting this analysis to estimate the accuracy of our predictions. We have now included a table of the top 10 most disordered proteins of known function (Supplementary file 1). We refer to this table in the text and note that our cluster-based annotations match the known functions for 5 of 10 of these proteins in subsection “Evolutionary signatures of function can be used for functional annotation of fully disordered proteins”.

5) The authors refer to the molecular features as "evolutionary signatures" but wouldn't "functional signatures" be a better term? The title also hints at this…

Although we show that the “evolutionary signatures” are associated with function, and we believe they contain functional information, we do not think they are necessarily “functional signatures”. The evolutionary signatures summarize the evolution of molecular features in the IDRs, and we show that they point to function. However, we do not think these should be called “functional signatures” directly. We also think that the current term “evolutionary signatures” emphasizes that these sequence-derived signatures are based on evolutionary information, a point that would be lost if we were to term them “functional signatures”.

6) The authors often use the term homology when they mean sequence similarity. They are not equivalent. For example, from subsection “Clustering of proteome-wide evolutionary signatures”:"For example, the cluster with the highest amount of "homologous" IDRs according to this threshold (top 1% homology) is cluster Q, with 8.9% homologous IDRs. However, the vast majority of the clusters have negligibly homologous IDRs; for example, 17/23 clusters have less than 1% homology between IDRs."It would be better to write:"For example, the cluster with the most similar IDRs according to this threshold (top 1% similarity) is cluster Q, with 8.9% sequence identity across the IDRs. However, the vast majority of the clusters have less similar IDRs; for example, 17/23 clusters have less than 1% sequence identity between IDRs."

We thank the reviewers for this comment. We have changed instances where we incorrectly used “homology” to “sequence similarity” (in the Results section as well as in the Materials and methods section), as the reviewers have suggested.

7) Following up on the entire paragraph from subsection “Clustering of proteome-wide evolutionary signatures” mentioned above, how are the sequences from different IDRs in the clusters compared. It says with Blosum62, but how are they aligned? Gap penalties? What information do pairwise distances between sequences that are this divergent provide?

We have included information about how the sequences are aligned (global alignment, with gap opening penalty of 0 and gap extension penalty of 1) in subsection under “Clustering of proteome-wide evolutionary signatures”. We agree with the reviewers that there is little information in the pairwise distances for divergent sequences, but this is precisely the point of this control experiment. We wanted to make sure that our clustering results could not be explained by sequence similarity, and distance-based metrics are the most commonly used method for measuring this property.

8) Disorder is predicted using DISOPRED3 on the S.cerevisiae proteome, but if disorder is only predicted for one species, how do you know that disorder is conserved for sequences in each alignment? Disorder is not necessarily conserved across sequences, not even across orthologs, see Montanari et al., 2011 and for further discussion see Ahrens et al2017.

The reviewers are correct that we only run DISOPRED3 on the S.cerevisiae proteome. Although there are exceptions, the presence of most disordered regions is conserved across orthologs. Furthermore, because we use a minimum of 10 (and often more than 15) orthologs in each alignment, even if the IDR is deleted in a few species, it will have a limited effect on our analysis. Nevertheless, we have now included our assumption that the presence of IDRs is conserved across orthologs in the Discussion section, and have cited Ahrens et al., 2017 and Montanari et al., 2011 in this section as well.

9) It is stated on numerous occasions that disordered regions are rapidly evolving and they most frequently do, but not always. Some disordered regions that also are predicted to have secondary structure to evolve slow, see Ahrens et al., 2018 and Ahrens et al., 2016.

The reviewers are correct that there are a minority of disordered regions that are slowly evolving. We have now included this point in the Discussion section and cited Ahrens et al., 2018 and Ahrens et al., 2016 (along with Brown et al., 2002).

10) Another recent paper that discusses functional constraint in disordered regions and due to its relevance, it ought to be referred to from the current manuscript, see Afanasyeva et al., 2018.

We have added this citation in the Introduction, where we include other studies on evolutionary constraint in disordered regions.

11) Figure 1—supplementary figure 2 shows nothing. It is supposed to show very little, but really, this shows nothing. If this is correct, can you add what the data looks like for ordered regions as a comparison?

We have added the non-IDR regions as a comparison, as suggested (Figure 1— figure supplement 2).